# New tools for automated high-resolution cryo-EM structure determination in RELION-3

Jasenko Zivanov[1†], Takanori Nakane[1†], Björn O Forsberg[2†], Dari Kimanius[2], Wim JH Hagen[3,4], Erik Lindahl[2,5]*, Sjors HW Scheres[1]*

[1]MRC Laboratory of Molecular Biology, Cambridge, United Kingdom; [2]Department of Biochemistry and Biophysics, Science for Life Laboratory, Stockholm University, Stockholm, Sweden; [3]Structural and Computational Biology Unit, European Molecular Biology Laboratory, Heidelberg, Germany; [4]Cryo-Electron Microscopy Service Platform, European Molecular Biology Laboratory, Heidelberg, Germany; [5]Department of Applied Physics, Swedish e-Science Research Center, KTH Royal Institute of Technology, Stockholm, Sweden

**Abstract** Here, we describe the third major release of RELION. CPU-based vector acceleration has been added in addition to GPU support, which provides flexibility in use of resources and avoids memory limitations. Reference-free autopicking with Laplacian-of-Gaussian filtering and execution of jobs from python allows non-interactive processing during acquisition, including 2D-classification, *de novo* model generation and 3D-classification. Per-particle refinement of CTF parameters and correction of estimated beam tilt provides higher resolution reconstructions when particles are at different heights in the ice, and/or coma-free alignment has not been optimal. Ewald sphere curvature correction improves resolution for large particles. We illustrate these developments with publicly available data sets: together with a Bayesian approach to beam-induced motion correction it leads to resolution improvements of 0.2–0.7 Å compared to previous RELION versions.
DOI: https://doi.org/10.7554/eLife.42166.001

*For correspondence:
erik.lindahl@scilifelab.se (EL);
scheres@mrc-lmb.cam.ac.uk
(SHWS)

†These authors contributed
equally to this work

Competing interest: See
page 18

Reviewing editor: Edward H
Egelman, University of Virginia,
United States

## Introduction

Macromolecular structure determination by single-particle analysis of electron cryo-microscopy (cryo-EM) images has undergone rapid progress in recent years. Before 2010, cryo-EM structures to resolutions beyond 4 Å had only been obtained for large icosahedral viruses, for example *Jiang et al., 2008*, *Yu et al., 2008*, and *Zhang et al., 2008*. Since then, cryo-EM structures to resolutions that are sufficient for de novo atomic modelling have been achieved for a wide range of samples (*Fernandez-Leiro and Scheres, 2016*), and several structures beyond 2 Å resolution have been reported (*Merk et al., 2016*; *Bartesaghi et al., 2018*). In 2012, the availability of the first prototypes of direct electron detectors, which recorded images with unprecedented signal-to-noise ratios (*McMullan et al., 2009*), represented a crucial step forward. This advance was closely followed and partially overlapped with important improvements in image processing software. Together, they led to what was termed the 'resolution revolution' in cryo-EM structure determination (*Kuhlbrandt, 2014*).

Since the start of this revolution, many new software solutions have been introduced. Some new programs exploited specific opportunities provided by the new detectors, like the correction of beam-induced motions through movie processing (*Brilot et al., 2012*; *Li et al., 2013*; *Rubinstein and Brubaker, 2015*; *Abrishami et al., 2015*; *McLeod et al., 2017*; *Zheng et al., 2017*).

Other programs tackled individual steps in the single-particle workflow, like fast and robust estimation of the contrast transfer function (CTF) (*Rohou and Grigorieff, 2015*; *Zhang, 2016*), local-resolution estimation (*Kucukelbir et al., 2014*), or on-the-fly processing of images while they are being acquired at the microscope (*de la Rosa-Trevín et al., 2016*; *Biyani et al., 2017*). In the mean time, also general-purpose programs that already existed, like EMAN2 (*Tang et al., 2007*), Frealign (*Grigorieff, 2007*) or IMAGIC (*van Heel et al., 1996*), were further improved and used for high-resolution structure determination. Both old and new programs benefitted from the increase in image signal-to-noise ratios provided by the new detectors.

The availability of the first direct-electron detectors coincided with the introduction of the RELION software (*Scheres, 2012b*). This computer program introduced an empirical Bayesian approach, in which optimal Fourier filters for alignment and reconstruction are derived from the data in a fully automated manner (*Scheres, 2012a*). This strongly reduced the need for user expertise in tuning *ad-hoc* filter parameters compared to existing programs, and probably contributed to the rapid uptake of cryo-EM structure determination by non-expert groups. Nowadays, RELION is widely used (*Patwardhan, 2017*), and similar statistical approaches have been implemented in software packages that were developed since: cryoSPARC (*Punjani et al., 2017*) uses the same regularised likelihood optimisation target as RELION, and SPHIRE (*Moriya et al., 2017*) and cisTEM (*Grant et al., 2018*) implement likelihood-based classification methods.

RELION underwent a major change in 2016 with the release of its second version, which comprised a pipelined approach for the entire single-particle workflow (*Fernandez-Leiro and Scheres, 2016*) and used acceleration on graphical proceing units (GPUs) to reduce its computational costs (*Kimanius et al., 2016*). In addition, its functionality was expanded with the incorporation of subtomogram averaging (*Bharat et al., 2015*) and helical reconstruction (*He and Scheres, 2017*). In this paper, we introduce version 3 of RELION. In the Materials and methods section, we describe algorithmic improvements and changes in implementation that have been made throughout the single-particle pipeline in RELION-3. In the Results and Discussion section, we illustrate how these changes allow for improved automation, fast accelerated reconstruction on normal hardware (CPUs) in addition to GPUs, and how the new features enable higher resolution reconstructions compared to previous releases.

## Materials and methods

### Key resources table

| Reagent type | Designation | Reference | Identifier |
| --- | --- | --- | --- |
| software | RELION | *Scheres, 2012b* | RRID:SCR_016274 |

### Laplacian-of-Gaussian auto-picking

The recommended procedure for the selection of particles from micrographs in previous versions of RELION was to calculate templates for reference-based auto-picking from particles that were manually picked in a subset of the micrographs (*Scheres, 2015*). To allow non-interactive, on-the-fly processing, we implemented a reference-free particle picking method based on the Laplacian-of-Gaussian (LoG) filter. Our method is related to the approach in DoGpicker (*Voss et al., 2009*), which uses a Difference-of-Gaussian (DoG) filter. Similar to the implementation in swarmPS (*Woolford et al., 2007*), we apply a LoG-filter with an optimum response for blobs of size $d$ by multiplying the Fourier transform of a micrograph by

$$\frac{|k|^2}{\sigma^2} \exp\left(\frac{|k|^2}{-2\sigma^2}\right) \tag{1}$$

where $k \in \mathbb{R}^2$ is the image frequency, $\sigma = 2/d$, and the division by $\sigma^2$ outside the exponentiation provides a normalised response of the filter for different values of $d$.

In order to deal with elongated particles, the user specifies a minimum ($d_{min}$) and maximum ($d_{max}$) particle diameter. The program then applies four LoG-filters to detect areas in the micrograph with blobs that are smaller than $d_{min}$ (with $d = d_{min}/n$ and $n = 2, 3, 4$ or $5$); four filters to detect blobs that

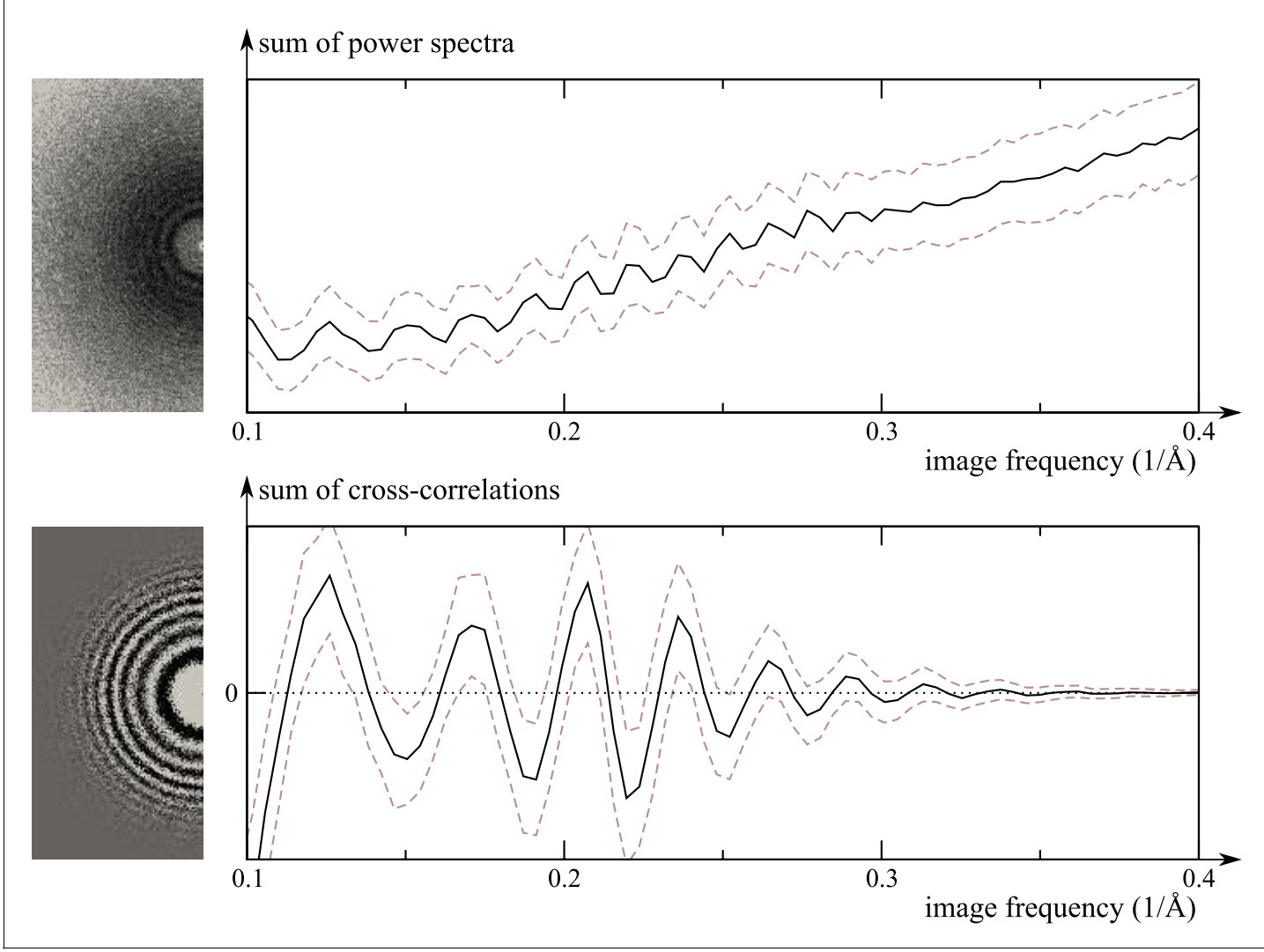

**Figure 1.** Comparison of the traditional power-spectrum-based CTF estimation technique with reference-map-based CTF refinement. Top: Sum over the power spectra $||X_k||$ of all polished particle images in micrograph EMD-2984_0000 of EMPIAR-10061 (left) and the averages and standard deviations of that sum over Fourier rings between 10 Å and 2.5 Å (right). The dashed grey curves are located one std. deviation of each ring above and below the average. The overall rise in spectral power as a function of frequency is a consequence of particle polishing. Bottom: The corresponding plot for the sum over the real components of $V_k^* X_k$, instead of the power spectra. Note that the latter term is not only centered around zero, obviating the need to estimate the background intensity, but it also offers a higher signal-to-noise ratio.
DOI: https://doi.org/10.7554/eLife.42166.002

are larger than $d_{max}$ (with $d = n * d_{max}$ and $n = 2, 3, 4$ or $5$); and three filters to detect blobs within the user-specified particle size (with $d = d_{min}$, $d = (d_{min} + d_{max})/2$ and $d = d_{max}$).

Only pixels for which the highest LoG-filtered value over all eleven filtered micrographs ($LoG_{best}$) is above a user-controlled threshold and the corresponding blob size $d_{best}$ is in the user-specified particle diameter range are considered for particle picking. Particles are picked iteratively at the highest remaining $LoG_{best}$ value, and around each picked particle one discards all pixels within a circle with diameter $d_{best}$. This process is repeated until no suitable pixels remain. To normalise the threshold for $LoG_{best}$ over many different data sets, the default threshold is set to the mean of all $LoG_{best}$ values for which the corresponding $d_{best}$ is in the user-specified particle diameter range, and deviations from the default are expressed in standard deviations of those same values.

## CTF refinement

In RELION, per-micrograph CTF parameters are determined through wrappers to CTFFIND (*Rohou and Grigorieff, 2015*) or Gctf (*Zhang, 2016*). These programs fit CTFs to the Thon rings visible in the power spectra of (patches of) micrographs. In RELION-3, we have implemented a program to refine the CTF parameters, that is to re-estimate defocus and astigmatism, using a 3D reference structure. This allows CTF estimation to exploit both the phases and the amplitudes of the experimental images, instead of having to rely exclusively on their power spectra. This approach thus produces significantly more reliable estimates, due to higher signal-to-noise ratios and also because it does not require separation of the Thon rings from the background intensity of the power spectrum. This is illustrated in *Figure 1*. The increased stability of the CTF fitting can be leveraged to estimate independent defoci for individual particles. Similar functionality also exists in Frealign (*Grigorieff, 2007*) and cisTEM (*Grant et al., 2018*).

Formally, CTF refinement in RELION-3 works as follows. Let $V$ be the (complex-valued) Fourier-space representation of the reference structure. We assume the viewing direction and particle position to be known for each observed particle $p$. This allows us to project and shift the 3D reference, yielding $V_p$, the corresponding reference projection in 2D. The Fourier space representation of experimental particle image $X_p$ is now assumed equal to $V_p$, multiplied by the CTF and distorted by noise. We could then estimate the optimal CTF from $V_p$ and $X_p$ by determining a set of CTF-parameters $\theta$ that minimises the squared difference between them:

$$\theta = \operatorname*{argmin}_{\theta'} \Big( \sum_{k,p} g_k (X_{p,k} - \mathrm{CTF}(\theta',k) V_{p,k})^2 \Big). \tag{2}$$

Instead, we maximise their correlation:

$$\theta = \operatorname*{argmax}_{\theta'} \Big( \sum_{k} g_k \mathrm{CTF}(\theta',k) \sum_{p} \mathrm{Re}(V_{p,k}^* X_{p,k}) \Big), \tag{3}$$

where the asterisk indicates complex conjugation, $k$ is the 2D Fourier-space frequency and per-frequency weights $g_k$ are given by the FSC between two independently refined half-maps – this a heuristic that serves to attenuate the effect of frequencies where the reference is less reliable.

The value of the CTF for a spatial frequency $k$ is defined as follows:

$$\mathrm{CTF}(k) = -\sin\Big( \pi\lambda \big(\delta_0 + \delta_A \cos(2(\phi_k - \phi_A))\big)|k|^2 + \frac{\pi}{2} C_s \lambda^3 |k|^4 - \chi \Big), \tag{4}$$

where $\lambda$ is the wavelength of the electrons, $\delta_0$ the average defocus, $\delta_A$ half the difference between the two effective defoci resulting from astigmatism, $\phi_k$ and $\phi_A$ are the azimuthal angles of $k$ and the major astigmatism axis, respectively, $C_s$ is the spherical aberration constant of the microscope and $\chi$ the overall phase shift resulting from amplitude contrast and, if one is used, a phase plate.

The expression in *Equation 3* allows the CTF-parameters $\theta = \{\delta_0, \delta_A, \phi_A, C_s, \chi\}$ to be determined either as constant for all particles in an entire micrograph or for every particle separately. In the latter case, the second sum consists of one single term corresponding to the index $p$ of the particle in question, and the result will be independent on the number of particles per micrograph. Our implementation allows for the estimation of $\delta_0$ on a per-particle basis, $\delta_A$ and $\phi_A$ on a per-particle or per-micrograph basis, and $C_s$ and $\chi$ on a per-micrograph basis. In our experiments, we have found that for many data sets the best results are obtained by estimating only $\delta_0$ for individual particles, while leaving all the other parameters unchanged.

Because the large number of CTF-parameters for a given dataset introduces the risk of overfitting, it is important to maintain a strict separation between the two independently refined half-sets. For that reason, the reference image $V_p$ used to estimate the parameters of particle $p$ is always obtained by projecting the reference map computed from the half-set to which particle $p$ belongs. For completeness, we note that by default RELION combines the lowest frequencies of the two half-set reconstructions in order to avoid a situation where the two refinements converge onto reconstructions with small differences in orientation. Since this combination is only done for frequencies below 40 Å, this does not have noticeable effects on the CTF refinement.

## Beam-tilt estimation and correction

A tilted beam alters the phase delays of the scattered beam components with respect to the unscattered reference beam. The delays $\psi_1$ and $\psi_2$ for a pair of corresponding Friedel mates are then no longer identical. We can decompose them into a symmetrical component $\psi_s = (\psi_1 + \psi_2)/2$ and an anti-symmetrical one, $\psi_a = (\psi_1 - \psi_2)/2$. The symmetrical part only induces astigmatism, which is already being modelled reasonably by our CTF model (*Equation 4*). The anti-symmetrical part, however, induces axial coma, that is a phase shift $\phi$ in the respective image frequencies. The magnitude of $\phi$ as a function of tilt $b \in \mathbb{R}^2$ is given by (*Glaeser et al., 2011*; *Zemlin et al., 1978*):

$$\phi(b) = 2\pi C_s \lambda^2 <k,b>|k|^2, \tag{5}$$

where $C_s$ is the spherical aberration constant of the microscope, $\lambda$ the wavelength of the electrons and $<,>$ denotes the scalar product in 2D.

We have developed an efficient method to determine the tilt vector $b$ from a set of observations of a particle and the corresponding high-resolution reference. Let $V_{j,k} \in \mathbb{C}$ denote the predicted complex amplitude of image frequency $k$ of particle $j$ (including the effects of the CTF) and $X_{j,k} \in \mathbb{C}$ the corresponding amplitude observed in the experimental image $X$. We aim to find a vector $b \in \mathbb{R}^2$ that minimises the squared difference between the observations and the prediction:

$$b = \underset{\beta}{\mathrm{argmin}} \sum_{j,k} |z_k(\beta) V_{j,k} - X_{j,k}|^2, \tag{6}$$

where the phase shift is expressed as a multiplication with a complex number: $z_k(\beta) = \exp(i\phi(\beta))$. Since the error in *Equation 6* is a sum of quadratic functions, the equation can be transformed into:

$$b = \underset{\beta}{\mathrm{argmin}} \sum_k w_k |z_k(\beta) - q_k|^2 \tag{7}$$

with the per-pixel optima $q_k \in \mathbb{C}$ and weights $w_k \in \mathbb{R}$ given by:

$$q_k = \sum_j (X_{j,k} V_{j,k}^*) / \sum_j |V_{j,k}|^2 \tag{8}$$

$$w_k = \sum_j |V_{j,k}|^2, \tag{9}$$

where the asterisk indicates complex conjugation. The computation of $q_k$ can be thought of as an averaging of the phase differences between the projected reference and the experimental images for Fourier-space pixel $k$ over all the images. Because of the cyclical nature of phase angles, a naive average over the angles themselves would not produce the desired result.

This formulation allows us to first compute $q_k$ and $w_k$ for each pixel by only iterating over the dataset once. This takes less than one hour for most datasets. Once the $q_k$ and $w_k$ are known, we can fit the optimal $b$ to them using an iterative non-linear method, such as Nelder-Mead downhill simplex (*Nelder and Mead, 1965*). Due to the small number of $k$, this iterative optimisation then only takes seconds.

In addition, inspecting the phases of $q_k$ allows us to ascertain that the effects observed are really those of a tilted beam. If we were to merely fit the optimal tilt vector $b$ to the data directly, we would run the risk of fitting $b$ to noise. Example plots of phase angles of $q_k$ are shown in *Figure 4—figure supplement 1*. Furthermore, such phase-error plots make it possible to observe the effects of higher-order anti-symmetrical aberrations, such as three-fold astigmatism (*Saxton, 1995*; *Krivanek and Stadelmann, 1995*). Although no methods are available to correct for these yet, we are currently investigating these.

Once the tilt vector $b$ is known, it can be used to account for the expected phase error in $X$ during refinement and reconstruction. This functionality has already been present in RELION since release 2.0, but no practical methods were available to estimate $b$ until now.

## Ewald sphere correction

We implemented the single-side band algorithm to correct for Ewald sphere curvature as proposed by (*Russo and Henderson, 2018*), which bears resemblance to the method implemented in Frealign

(*Grigorieff, 2007*; *Wolf et al., 2006*). Our implementation was made in the `relion_reconstruct` program, which was also parallelised using the message-passing interface (MPI) standard.

Briefly, one divides the Fourier transform of each extracted particle into two halves and multiplies each half with one of the following single-side band CTFs:

$$
\begin{aligned}
CTFP &= e^{+i(\psi+\pi/2)} \text{and} \\
CTFQ &= e^{-i(\psi+\pi/2)},
\end{aligned}
\tag{10}
$$

where $\psi$ is the phase shift of the electron wave caused by the objective lens, and the term $\pi/2$ is the phase shift upon scattering by the specimen. These complex multiplications leave the amplitudes of the Fourier components intact, but change the phases, resulting in a breakage of Friedel symmetry. To avoid discontinuities at the edge between the two halves of the Fourier transform, the multiplication is done in sectors by rotating the edge between the two halves. Two Fourier transforms, one for each side-band, are then obtained for each particle by combination of those parts of the Fourier transform that are furthest away from the edge in the different sectors. The number of sectors can be controlled by the user. By default it is set to two.

The two Fourier transforms of each particle sample the 3D transform in positions that are slightly above and below the central section. Both are back-transformed, masked in real space to remove noise in the area around the particle, and then transformed back into Fourier space before being inserted into their correct position in the 3D transform. The user has the option to use either upward or downward curvature of the Ewald sphere. The distance $\Delta z^*$ from the central section is calculated as:

$$
\Delta z^* = \lambda/2|k|^2.
\tag{11}
$$

Because an unknown number of mirroring operations may have been introduced during the data acquisition and image processing, the actual direction of curvature of the Ewald sphere is often unknown for a given data set. Therefore, the user may typically want to perform Ewald sphere correction with both directions. For this purpose, we implemented the `-reverse_curvature` option. If Ewald sphere curvature is a limiting factor for the resolution of a reconstruction, one direction of curvature should yield a higher resolution map than the other. *DeRosier (2000)* estimates that the frequency $k^*$ for which Ewald sphere curvature becomes limiting for a particle with diameter $d$ is approximated by:

$$
k^* = \sqrt{\frac{1.4}{d\lambda}}.
\tag{12}
$$

## CPU vector acceleration

The large computational load of the cryo-EM pipeline requires RELION to be highly parallel. In RELION-2, a new code-path was introduced, which off-loaded the core computations onto graphical processing units (GPUs) (*Kimanius et al., 2016*). This new path was by design less demanding of double precision arithmetics and memory, and provided a large improvement in data throughput by virtue of the greatly increased speed. However, the GPU acceleration is only available for cards from a single vendor, and it cannot use many of the large-scale computational resources available in existing centres, local clusters, or even researchers' laptops. In addition, the memory available on typical GPUs limits the box sizes that can be used, which could turn into a severe bottleneck for large particles. For RELION-3, we have developed a new general code path where CPU algorithms have been rewritten to mirror the GPU acceleration, which provides *dual* execution branches of the code that work very efficiently both on accelerators as well as the single-instruction, multiple-data vector units present on traditional CPUs. This has enabled lower precision arithmetics, it reduces memory requirements considerably, and it also makes it possible to exploit the very large amounts of memory that can be fitted on CPU servers. In addition, it provides a transparent source code where new algorithms may be accelerated immediately, and it enables a significant speedup compared to the *legacy* code path, which is nonetheless still present in RELION-3 for comparative purposes.

## Python scripts for automated processing

As of release 2.0, RELION has used a pipelined approach, where the execution of individual command-line programs is organised into *jobs*. Multiple different types of jobs exist for the different tasks in the single-particle workflow, for example there are specific jobs for CTF estimation, particle picking, 2D classification and 3D refinement. Each job takes specific types of files, called *nodes* as input and produces other nodes as output. By using output nodes from one job as input for the next job, the entire process of structure determination is expressed as a directional graph. Parameters for jobs are typically provided through the GUI, where jobs can be executed or scheduled for later, possibly repeated, execution.

In RELION-3, we have expanded the functionality to schedule and execute jobs from the GUI to the command line by modifications in the `relion_pipeliner` program. This has allowed the generation of (python) scripts to perform complicated, standardised tasks by execution of a series of jobs. All jobs executed by these scripts become part of the standard pipeline, and one can use the GUI to follow the process live, or to do other things in the same directory meanwhile. With RELION-3, we distribute two example scripts. We envision that these will serve as inspiration for scripts written by others to reflect their specific needs.

The first example script is called `bfactor_plot.py`. It automatically runs multiple 3D refinements on random subsets of an input data set, where each time the size of the subset is doubled from a user-specified minimum number of particles. The script then automatically generated a B-factor plot according to Rosenthal and Henderson (*Rosenthal and Henderson, 2003*), where $1/d^2$, with $d$ being the resolution of each refinement, is plotted against the logarithm of the number of particles in the subset. The script also calculates the B-factor from the slope of a fitted line through the plotted points, and uses extrapolation of this line to make predictions about the number of particles needed to achieve higher resolutions.

The second example script is called `relion_it.py`. It is designed to provide on-the-fly feedback on the quality of micrographs, while they are being acquired at the microscope. The script repeatedly executes jobs for micrograph-based motion correction, CTF estimation, LoG-based auto-picking and particle extraction. In the mean time, the script keeps track of how many particles have been extracted, and can also execute 2D and/or 3D classification jobs in batches of a user-defined number of particles. If no 3D reference is available for 3D classification, an SGD initial model generation job will be executed to create one automatically. In addition, it is possible to replace the LoG-based auto-picking job by a template-based auto-picking job once a suitable 3D reference has been obtained from 3D classification of the LoG-picked particles.

Although almost all parameters from the GUI are accessible through the `relion_it.py` script, the procedure has been designed to rely on only a few dataset-specific parameters. Besides parameters describing the microscopy and computational setup, the most important parameters are $d_{min}$ and $d_{max}$ for the LoG-based auto-picking and box sizes for the extracted particles. The default thresholds for the LoG-based and template-based auto-picking jobs are set to relatively low values. This typically results in many false positives in the picking, which are then sorted out in the subsequent classifications. This way, completely non-interactive, on-the-fly feedback on data quality may be achieved.

## Other developments

Besides the developments described above, RELION-3 comprises two major algorithmic advances that have already been described elsewhere.

Firstly, Bayesian particle polishing implements a Gaussian Process regression algorithm for estimating beam-induced motion tracks for individual particles, and an improved B-factor estimation algorithm for resolution-dependent weighting of individual movie frames. Although not strictly necessary, Bayesian polishing is typically performed after motion-correction at the micrograph (patch) level by MotionCor2 (*Zheng et al., 2017*). To expose the metadata of this algorithm better to the Bayesian polishing approach, we also implemented our own version of the MotionCor2 algorithm. Unlike the UCSF implementation, our version does not use GPU-acceleration but runs on multi-core CPUs using OpenMP multi-threading. For $4k \times 4k$ movies, and using 12 cores, our program runs at comparable speeds to the UCSF implementation. More details about the Bayesian polishing and our implementation of the MotionCor2 algorithm have been described by *Zivanov et al, (2018)*.

Secondly, multi-body refinement implements an automated and iterative approach for focused refinement with partial signal subtraction on multiple independently moving parts of a complex (*Nakane et al., 2018*). This approach is useful for improving densities of flexible multi-domain complexes, and yields insights into the type of inter-domain motions that are present in those.

In addition, many smaller programs and functionalities have been added throughout the single-particle processing workflow. We highlight the following. The motion correction, CTF estimation and auto-picking jobs now output PDF files with metadata plots for all micrographs. We have improved our 3D initial model generation by adopting a stochastic gradient descent (SGD) procedure that follows the implementation in cryoSPARC much more closely than our previous version. We achieved considerably speedups in 2D and 3D classification by using subsets of particles in the first few iterations, which was inspired by a similar implementation in cistem (*Grant et al., 2018*). We have added a program called `relion_star_handler` that implements a range of useful operations for metadata files in the star format (*Hall, 1991*). And we have implemented a program called `relion_align_symmetry` that aligns the symmetry axes of a 3D map in arbitrary orientation according to the conventions in RELION.

## Results and discussion

### CPU vector acceleration

The functions that perform the expectation step (particle alignment) in RELION-3 can now use the same code path and structure as the GPU-accelerated code that was introduced in RELION-2, even if executing on systems without GPUs. This has led to several key improvements. First, the algorithm strength reductions first introduced for the GPU code makes it possible to use single precision in most places, which cuts the memory footprint in half for large parts of the code. Second, data has been organised to better fit modern hardware. The legacy CPU code would for instance allocate space for all *possible* orientations under the current sampling level, while the new CPU code makes use of coalesced memory allocation limited to the orientations that will be investigated. These type of changes result in a higher compute intensity with more coalesced memory access in smaller allocations. It is evident that the generalised and more streamlined code path facilitates compiler optimisation compared to the previous CPU version. The standard 3D benchmark for example runs 1.5x and 2.5x faster on the previous (Broadwell) and current (Skylake) generations of x86 processors, respectively (*Figure 2A,B*), without compromising the results (*Figure 2—figure supplement 1*). However, the new coalesced memory allocation also makes it possible to instruct the compiler to generate efficient code targeting the single-instruction, multiple-data (SIMD) vector units present in modern CPUs. These operate on many input data elements at once, somewhat similar to GPUs, and typically also provide better throughput for single-precision data. Examples of this includes the so-called avx (advanced vector extensions) instructions present in all 64-bit x86 CPUs that operate on eight single-precision floating-point numbers, or the avx512 instructions present in the newest CPUs that extend the width to 16 single-precision numbers. By explicitly instructing the compiler to use avx-instructions for the new generalised code path, we see a further performance increase of 5.4x on Skylake-generation CPUs (*Figure 2B*). This enables x86 CPU nodes to approach the cost-efficiency of professional GPU hardware (*Figure 2C*), with results that are qualitatively identical to the legacy code path (*Figure 2—figure supplement 1*).

Notably, use of AVX-instructions does not improve on performance for legacy CPU-execution at all since the code parallelism is not expressed clearly enough for the compiler to recognise it. The performance also depends on which compiler is used, as they are able to optimise the program to varying degrees. The GNU compiler (GCC, verison 7.3) seems less able to convert RELIONs current formulation to efficient parallel instructions than the Intel compiler (ICC, version 2018.3.222), and presently the former provides no additional benefit on a Broadwell-generation CPU when the slightly newer AVX2 instructions are enabled. The better performance of icc is not unreasonable given that these benchmarks target Intel CPUs, although the negligible benefit of enabling AVX2 support in GCC is a bit surprising - hopefully this will improve in future compiler versions.

A handful of things are worth emphasising:

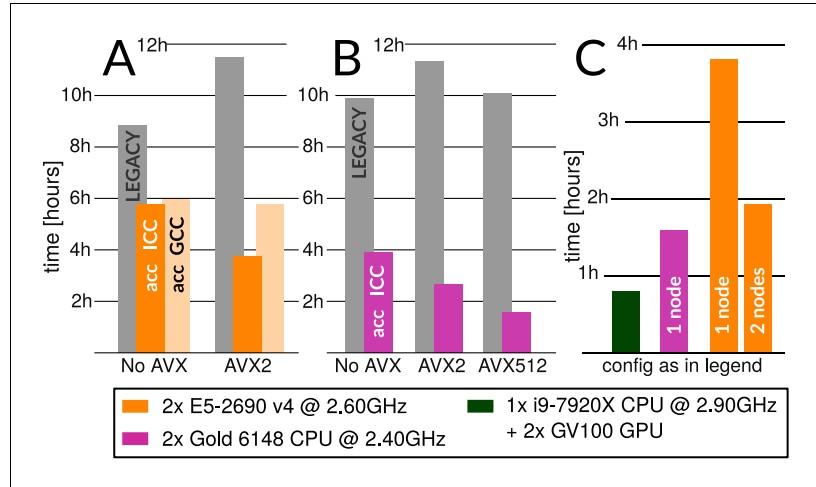

**Figure 2.** Accelerated CPU performance (**A**) Even when specific vector instructions are disabled, RELION-3 runs faster than RELION-2 even on the previous-generation Broadwell processors that are ubiquitous in many cryo-EM clusters worldwide. Enabling vectorisation during compilation with the Intel compiler benefits the new streamlined code path, but not the legacy code. (**B**) For latest-generation Skylake CPUs, the difference is much larger even with only AVX2 vectorisation enabled, and when enabling the new AVX512 instructions the performance is roughly 4.5x higher than the legacy code path. (**C**) The accelerated CPU code executing on dual-socket x86 nodes provides cost-efficiency that is at least approaching that of professional-class GPU hardware (but not consumer GPUs).

DOI: https://doi.org/10.7554/eLife.42166.003

The following figure supplement is available for figure 2:

**Figure supplement 1.** FCSs comparing half-sets in each type of run, legacy-CPU, acc-CPU, and acc-GPU, all reach the same resolution and sampling accuracy.

DOI: https://doi.org/10.7554/eLife.42166.004

- While the code has been streamlined to allow compilers to generate efficient x86 SIMD vector instructions, only preprocessor directives are used to aid the acceleration, and the new code path is therefore portable to any architecture.
- Since the vector acceleration executes on the CPU, it is straightforward to expand memory and use arbitrarily large box sizes (in contrast to the limitations present when offloading to (GPU) accelerators).
- For now, the legacy CPU execution is preserved for comparison. Legacy execution can be run with any relion_refine program by omitting the –gpu and/or –cpu flag.
- Due to complex compiler dependencies for GPU support, it might not be possible to enable simultaneous CPU and GPU acceleration support. Similarly, since the CPU acceleration targets the same routines as the GPU acceleration, it will not *per se* alleviate any CPU bottlenecks during GPU execution. The generalised path means RELION will branch into functions that launch highly parallel *either* on the CPU or GPU.
- Owing to the higher performance achieved through use of the non-free icc compiler, and the fact that current gcc versions do not yet automatically generate AVX512 instructions, we will provide pre-build RELION-3 binaries optimised for the latest CPUs in addition to source code distribution.

## Per-particle defocus correction

Per-particle defocus variations within a micrograph arise from different particles being situated at different heights within the field of view. This may happen when the ice layer is relatively thick, and/or when the ice layer is not perpendicular to the electron beam. The latter may occur when the sample support film wrinkles upon flash freezing or when the sample stage is tilted in the microscope, for example to ameliorate a problem of preferred particle orientations (*Tan et al., 2017*; *Naydenova and Russo, 2017*). As an extreme case of non-horizontal samples, we re-processed an influenza virus hemagglutinin dataset (EMPIAR-10097) that was collected at 40 degree stage tilt to

overcome a strong preferred orientation of the particles (*Tan et al., 2017*). At this tilt angle, the height differences within a single micrograph are more than 4000 Å. A reconstruction from this data set was originally published at 4.2 Å resolution. Recently, the same data were also re-processed in Warp and cryoSPARC, resulting in an improved resolution of 3.2 Å (*Tegunov and Cramer, 2018*).

We aligned and dose-weighted the 668 movies with our own implementation of the MotionCor2 algorithm, and estimated per-micrograph defocus parameters using CTFFIND4 (*Rohou and Grigorieff, 2015*). Because of the high tilt, Thon rings were very weak and the estimated resolution from CTFFIND4 was limited to about 10 Å. We applied LoG autopicking to an initial subset of 210 micrographs, and subjected the resulting 75,360 particles to 2D classification. We then used four good 2D class averages as templates for reference-based autopicking of all micrographs. The resulting 608,919 particles were then classified into 100 2D classes, from which we selected 361,507 particles that contributed to reasonable 2D class averages. We generated an initial 3D model using the SGD algorithm with C3 symmetry. Up to this point, we did not perform CTF-amplitude correction at resolutions lower than their first peak. A first 3D refinement with full CTF correction yielded only 6.6 Å resolution, and the FSC curve between the two half-maps showed strong oscillations, indicating inaccuracies in the CTF correction (*Figure 3A,B*). We then refined the per-particle defoci with a search range of ±2500 Å, which improved the resolution to 4.4 Å. Repeating per-particle defocus refinement and 3D refinement two more times yielded 4.0 and 3.8 Å. Since a fourth iteration did not improve the resolution any further, we then proceeded with Bayesian polishing, which improved the resolution to 3.5 Å. We then used 3D classification into six classes to select a subset of 152,960 particles contributing to the best class, again resulting in 3.5 Å resolution after 3D refinement. CTF refinement with beam-tilt correction and 3D refinement were then again iterated three times, giving 3.3 Å, 3.2 Å, and finally 3.1 Å resolution. During these refinements, 1364 particles were translated within 30 Å of their closest neighbour. Since duplicated particles may invalidate gold-standard

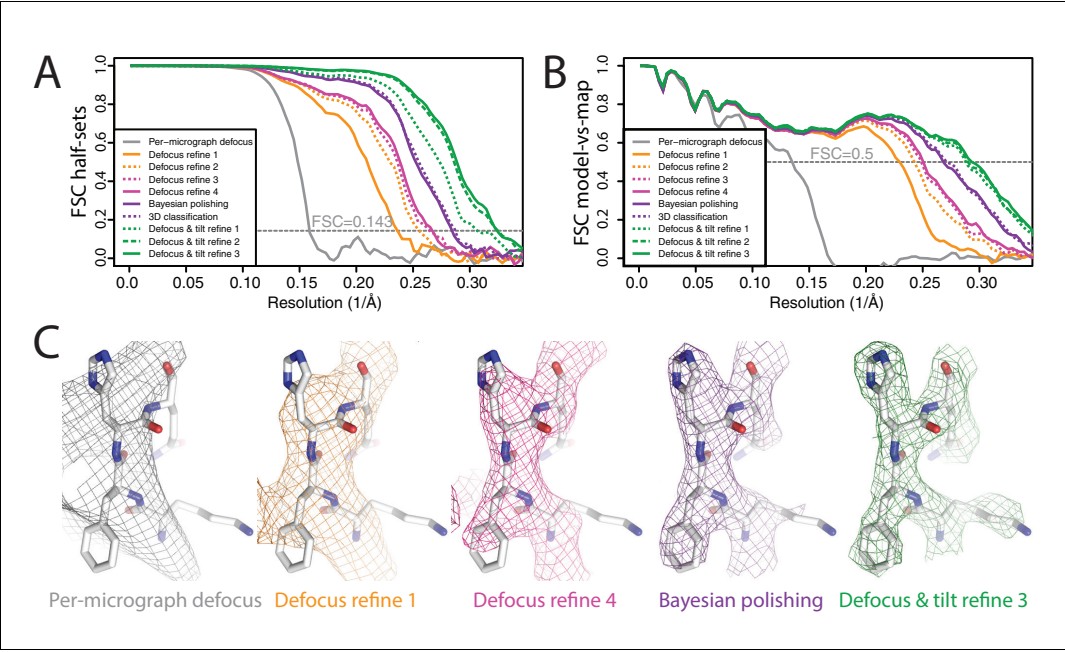

**Figure 3.** Per-particle defocus correction (**A**) FSC curves between independently refined half-maps for the different stages of processing as explained in the main text. (**B**) As in A, but FSC curves are between the cryo-EM maps and the corresponding atomic model (PDB-4FNK) (*Ekiert et al., 2012*). (**C**) Representative density features for some of the maps for which FSC curves are shown in A and B.

DOI: https://doi.org/10.7554/eLife.42166.005

The following figure supplement is available for figure 3:

**Figure supplement 1.** Per-particle defocus estimates (along Z) are plotted against the X,Y-coordinates of the particles in a representative micrograph.

DOI: https://doi.org/10.7554/eLife.42166.006

separation and thereby inflate resolution, we removed these duplicate particles before the final 3D refinement using new functionality in the 'Subset selection' jobtype on the GUI. Plots of the final defocus values for particles within a single micrograph clearly replicate the tilt geometry of the experiment (*Figure 3—figure supplement 1*).

This example demonstrates that even for models with intermediate resolution (in this case 6.6 Å) our per-particle defocus refinement can yield improved defoci. It also illustrates the robustness of the method against the initial values being far off. Using initial defoci that are closer to the correct ones will reduce the need for performing multiple cycles of per-particle defocus refinement. Therefore, in practice it would probably be faster to start from defoci provided by local CTF estimation programs, such as CTFTILT (*Mindell and Grigorieff, 2003*), Gctf (*Zhang, 2016*) or Warp (*Tegunov and Cramer, 2018*). Currently, this requires some scripting by the user, as the wrappers to CTFFIND4 and Gctf on the RELION-3 GUI do not allow local CTF estimation.

## Beam tilt correction

Until to date, beam tilt has been largely ignored in cryo-EM single-particle analysis. Prior to the resolution revolution, the amount of beam tilt present in typical data sets did not affect their achievable resolution to a noticeable extent. More recently, even though methods to correct for beam tilt already existed in RELION and Frealign, the lack of reliable tools to estimate beam tilt meant these corrections were hardly ever used. Beam tilt results from sub-optimal coma-free alignment of the microscope. Based on our experience, we estimate that currently available microscope alignment procedures may still leave up to a quarter mrad of beam tilt in the data. This will not limit the resolution of 3 Å structures, but as shown for $\beta$-galactosidase in section 3.6, such amounts of beam tilt are limiting for 2 Å structures. Therefore, as resolutions keep improving, beam tilt correction will gain in importance.

There are data acquisition scenarios where beam tilt becomes a limiting factor at much lower resolutions. With the elevated costs of high-end electron microscopes, faster data acquisition methods are highly sought after. A rate-limiting step in many data acquisition schemes is to physically move the sample stage from one hole in the carbon to the next, as one needs to wait for mechanical stage drift to settle. Because it is much faster to shift the electron beam instead, one could achieve higher data acquisition rates by using beam-shift to image multiple holes before shifting the stage. Beam-shift is already used extensively to take multiple images within a single hole, but because it introduces beam tilt, its use for imaging multiple holes has only recently attracted attention (*Cheng et al., 2018*). Although it is possible to actively compensate for the introduced beam tilt using the microscope's deflection coils (*Eades, 2006*; *Glaeser et al., 2011*), this procedure puts more stringent requirements on microscope alignment.

We evaluated our methods for beam tilt estimation and correction using two data sets on horse spleen apo-ferritin (purchased from Sigma), which were acquired with or without active beam tilt compensation. The data were recorded using serialEM (*Mastronarde, 2005*) in combination with a patch in the Thermo Fischer microscope software to perform active beam tilt compensation. (At the time of our experiment, active beam-tilt compensation was not yet implemented in serialEM.) For both data sets, we used the same R2/2 Quantifoil grid; the same Titan Krios microscope with a Gatan energy filter (with a slit width of 20 eV); the same K2-summit detector; and we used beam-shift to take five images in nine holes each before shifting the stage. This procedure was repeated nine times for the first data set and ten times for the second, because the area imaged in the latter contained fewer particles. We only used active beam tilt compensation for the first data set. Movies with a total dose of 41 electrons per Å$^2$ were fractionated into 40 frames with a pixel size of 1.04 Å. Template-based auto-picking yielded 104,992 particles for the first and 109,646 particles for the second data set. After selection of suitable 2D classes, the first and second data sets comprised 79,985 and 65,131 particles, respectively.

The first data set, with active beam tilt compensation, yielded a resolution of 2.3 Å after standard 3D auto-refinement, and 2.2 Å after Bayesian polishing and a second round of refinement (*Figure 4A,B*). The second data set, without active beam tilt compensation, yielded 2.9 Å after the first refinement, and 2.8 Å after Bayesian polishing. Beam tilt estimation was then performed independently for nine subsets of the data: one subset for each of the nine holes that were imaged before shifting the stage, and each subset thus comprising $10 \times 5$ movies.

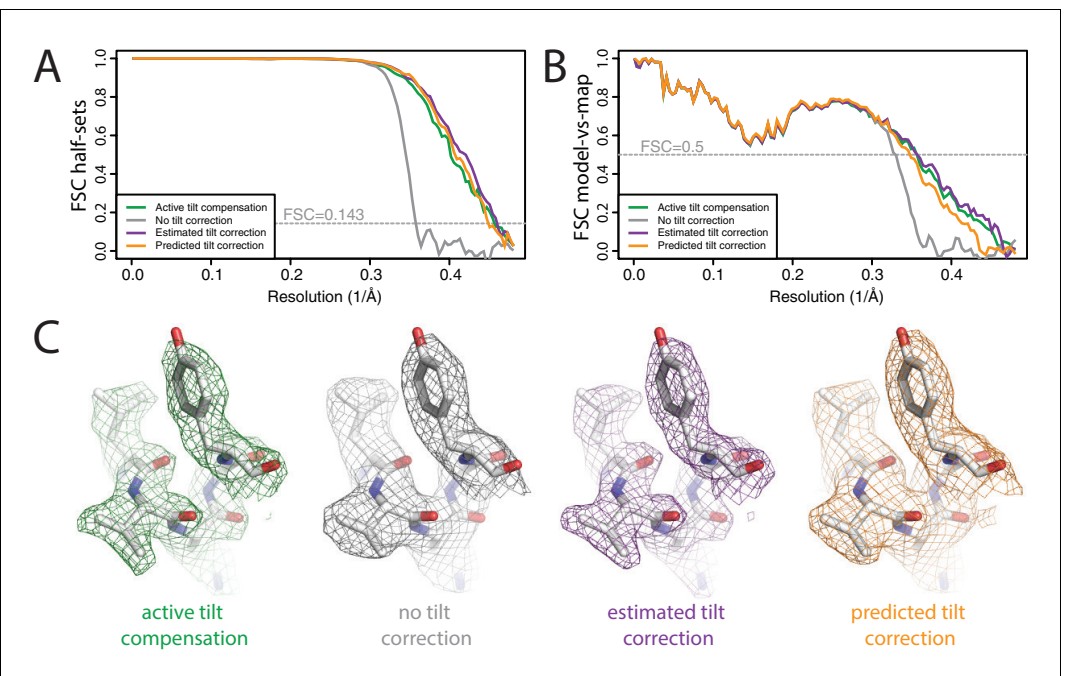

**Figure 4.** Beam tilt correction (**A**) FSC curves between independently refined half-maps for the different stages of processing as explained in the main text. (**B**) As in A, but FSC curves are between the cryo-EM maps and the corresponding atomic model (PDB-2W0O) (*de Val et al., 2012*). (**C**) Representative density features from the 2.2 Å map from the data set with active beam tilt compensation (left); the 2.9 Å map from the data set without active beam tilt compensation and without beam tilt correction (middle); and the 2.2 Å map from the data set without active beam tilt compensation but with beam tilt correction (right).

DOI: https://doi.org/10.7554/eLife.42166.007

The following figure supplements are available for figure 4:

**Figure supplement 1.** The average per-pixel phase differences (in radiants) between reference projections and the individual experimental particle images (i.e. the phase angle of $q_k$ in *Equation 8*).
DOI: https://doi.org/10.7554/eLife.42166.008

**Figure supplement 2.** The estimated beam tilt is plotted against the beam-shift values from serialEM in the X-direction (black circles) and in the Y-direction (red circles).
DOI: https://doi.org/10.7554/eLife.42166.009

Plots of the phase differences measured for each of the nine holes clearly show the effects of beam tilt at higher resolution (*Figure 4—figure supplement 1*). Smaller phase shifts in the opposite direction at intermediate resolution indicate that during refinement, the images have shifted off-center in an attempt to compensate for the beam tilt-induced phase shifts. A third round of refinement with correction for the estimated beam tilts improved the resolution to 2.2 Å. Plots of the estimated beam tilt against the beam-shift recorded by serialEM (*Figure 4—figure supplement 2*) revealed a linear relationship of 0.19 mrad of beam tilt per $\mu$m of beam-shift in both the X and Y directions. We then also ran a fourth refinement, where we predicted the beam tilt for each movie separately based on the recorded beam-shift values by serial EM. This resulted in a resolution of 2.2 Å. Features in the reconstructed density confirmed the improvement in resolution caused by beam tilt correction (*Figure 4C*). These results illustrate that beam tilt estimation and correction in RELION-3 may be a feasible alternative to active beam tilt compensation at the microscope. Future studies will need to explore which of these methods is preferable under different experimental conditions.

## Ewald sphere correction

Ewald sphere correction has also been largely ignored, although a few contributions have recently described high-resolution reconstructions where Ewald sphere correction did make a difference (*Tan et al., 2018*; *Zhu et al., 2018*). To test our Ewald-sphere correction implementation, we used

particle images of the P22 virion capsid protein (*Hryc et al., 2017*), which we downloaded from EMPIAR-10083 (*Iudin et al., 2016*). The diameter of the P22 virion is approximately 700 Å, which leads to an estimated resolution limit of 3.1 Å due to Ewald sphere curvature (*DeRosier, 2000*). To reduce memory consumption and to accelerate the processing, we down-sampled the particles to 1.54 Å/pixel, and skipped Fourier padding during refinement. As we did not have access to the original micrographs, we used the defocus parameters for 45,150 particles as they were refined by JSPR in the original publication, which were kindly provided by Wah Chiu. We ignored variations in magnification as estimated by JSPR, because magnification correction is not implemented in RELION-3. Standard 3D auto-refinement, with correction for an estimated beam tilt of 0.6 mrad in X and −0.13 mrad in Y, gave a reconstruction with an estimated resolution of 3.5 Å (*Figure 5A,B*). Subsequent Ewald sphere correction improved this to 3.3 Å. Features in the reconstructed density confirmed the improvement in resolution caused by beam tilt correction (*Figure 5C*). Ewald sphere correction with the opposite curvature gave only 3.9 Å. In theory, if one knows the number of mirroring operations during data acquisition and image processing, Ewald sphere correction could therefore also be used to determine the absolute hand of a structure. However, at the resolutions where Ewald sphere correction matters, this is often no longer a relevant question.

## Non-interactive data exploration

To illustrate the general applicability of the `relion_it.py` script, we re-processed our previously published γ-secretase data set (*Bai et al., 2015b*; *Bai et al., 2015a*), which we deposited at EMPIAR-10194. We chose this data set as a representative of a non-straightforward data set, as γ-secretase is a small (130 kDa ordered mass), asymmetrical, membrane-embedded protein complex.

Using default parameters, we performed motion correction of the 2,925 16-frame movies in our own implementation of the MotionCor2 algorithm, and estimated CTF parameters in Gctf (*Zhang, 2016*). LoG-based picking (with $d_{min} = 120$ and $d_{min} = 180$) at the default threshold led to 2.6 million particles (*Figure 6A*), which were divided in batches of 10,000 particles. Only the first batch was used for 2D classification, which revealed the presence of different views of the complex (*Figure 6B*). The script could have run similar 2D classification on all batches, for example to monitor data quality during data acquisition.

Instead, we chose to let the script proceed with SGD initial model generation and 3D classification on the same batch of 10,000 particles (*Figure 6C*). After 3D classification, the script automatically selects the 3D class with the highest resolution reconstruction. As all four classes had an equal resolution of 23 Å, the script selected the largest class (with an accumulated weight of 34.5%). However, the second largest class (with an accumulated weight of 34.4%) was actually the correct structure, which was obvious from a visual inspection of the density (*Figure 6C*). At this stage, we intervened. We stopped the script and edited the file `RELION_IT_2NDPASS_3DREF` to point to the correct map. This intervention probably reflects the weakest point in our otherwise non-interactive procedure. Robust, non-interactive selection of 3D class reconstructions (or 2D class averages) is a subject of ongoing research in our groups.

Upon re-starting the script, template-based auto-picking with the correct 3D structure and the default threshold (0.4) led to 1.8 million particles, which were then used for 2D classification into 200 classes. At this point, the non-interactive script finished and the rest of the data processing was performed manually. Selection of the best 2D class averages led to a subset comprising 665,102 particles, which were subjected to Bayesian particle polishing and two rounds of 3D classification into eight classes. Two classes, comprising 333,377 particles, were selected for refinement of the per-particle defocus values and separate estimations of the beam tilt in the six microscopy sessions that this data set comprises. A second 3D classification yielded a majority class with 240,777 particles, which led to a final resolution of 3.3 Å (*Figure 6E–G*).

When we originally processed this data set in RELION-1.3, we obtained a map at 3.4 Å resolution from 159,549 particles, which were selected after 3D classification of 412,272 polished particles (*Bai et al., 2015b*). This set of particles was obtained through careful tuning of template-based auto-picking parameters for each of the six microscopy sessions, and careful selection of 2D and 3D classes. To assess whether the non-interactive pre-processing procedure can compete with this expert-driven approach, we used the same set of 412,272 particles also for Bayesian polishing, refinement of per-particle defocus values, beam tilt estimation and 3D classification. The resulting selection of 246,602 particles gave a final resolution of 3.2 Å (*Figure 6E–G*). The improvement from

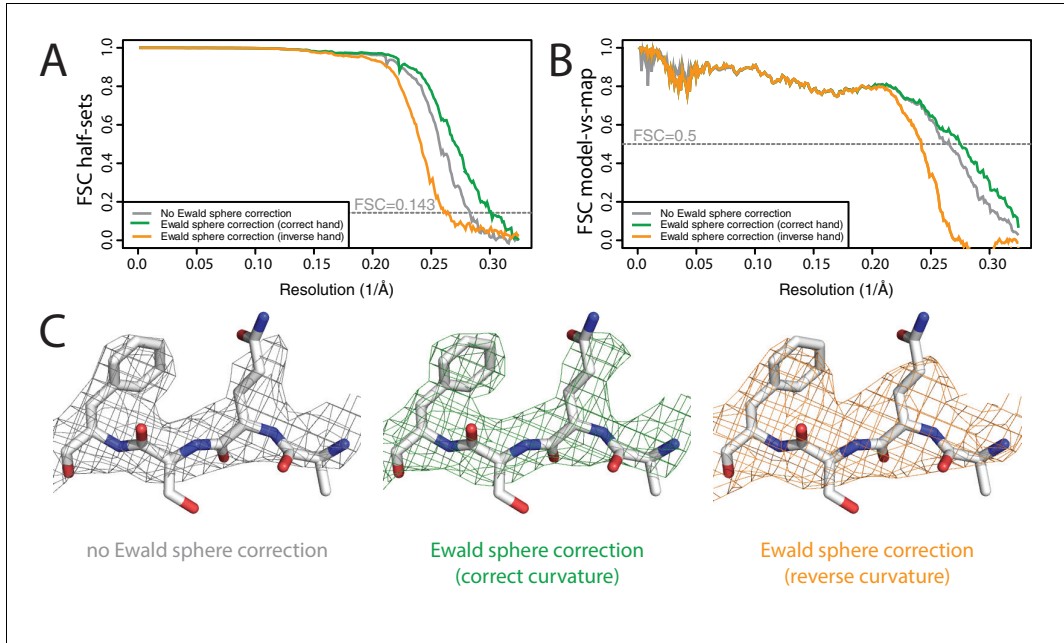

**Figure 5.** Ewald sphere correction (A) FSC curves between independently refined half-maps without Ewald sphere correction (grey); with Ewald sphere correction with the correct curvature (green) and with Ewald sphere correction with the inverse curvature (orange). (B) As in A, but FSC curves are between the cryo-EM and the corresponding atomic model (PDB-5UU5) (*Hryc et al., 2017*). (C) Representative density features without Ewald sphere correction (grey) with Ewald sphere correction and the correct curvature (green) and with Ewald sphere correction and the reverse curvature (orange).

DOI: https://doi.org/10.7554/eLife.42166.010

3.4 Å to 3.2 Å represents the contribution of the new functionality in RELION-3, whereas the gain from 3.3 Å to 3.2 Å represents the added value of the expert user. However, the map from the automatically pre-processed data does not correlate with the atomic model as well as one would expect from the half-map FSC curves. This suggests a higher fraction of false positives in the data resulting from the `relion_it.py` script compared to the expert selection, which may be caused by the relatively low default for the auto-picking threshold in the script. These results illustrate that non-interactive pre-processing gets close to the performance of a human expert, but for challenging data sets there is still a benefit in careful tuning of parameters. Future investigations will aim to reduce this gap.

## High-resolution refinement I: $\beta$-galactosidase

To illustrate the combined effects of the RELION-3 implementation on high-resolution structure determination, we re-processed the $\beta$-galactosidase data set at EMPIAR-10061 (*Bartesaghi et al., 2015*) that we also used to illustrate RELION-2.0 (*Kimanius et al., 2016*). This data set was made available to the community by the Subramaniam group. Using RELION-2.0, we previously obtained a 2.2 Å resolution reconstruction (EMD-4116).

We aligned the original, super-resolution movies using the UCSF implementation of MotionCor2 (version 1.0.5) (*Zheng et al., 2017*) in four by four patches with dose weighting and two-fold binning. CTF parameters were estimated using CTFFIND 4.1.10 (*Rohou and Grigorieff, 2015*). We initially picked 4084 particles using the LoG-based algorithm from a subset of 462 movies and classified these particles into 20 2D classes. Six of the resulting 2D class averages were used for template-based auto-picking of 229,553 particles from the entire data set. After an initial 2D classification into 200 classes, we chose to reject only the most obvious outliers, resulting in a data set of 217,752 particles. Although many of the selected particles were probably sub-optimal, we chose to keep them until the Bayesian polishing step, where relatively high particle numbers per micrograph may improve the description of the motion tracks. An initial 3D auto-refinement with these particles

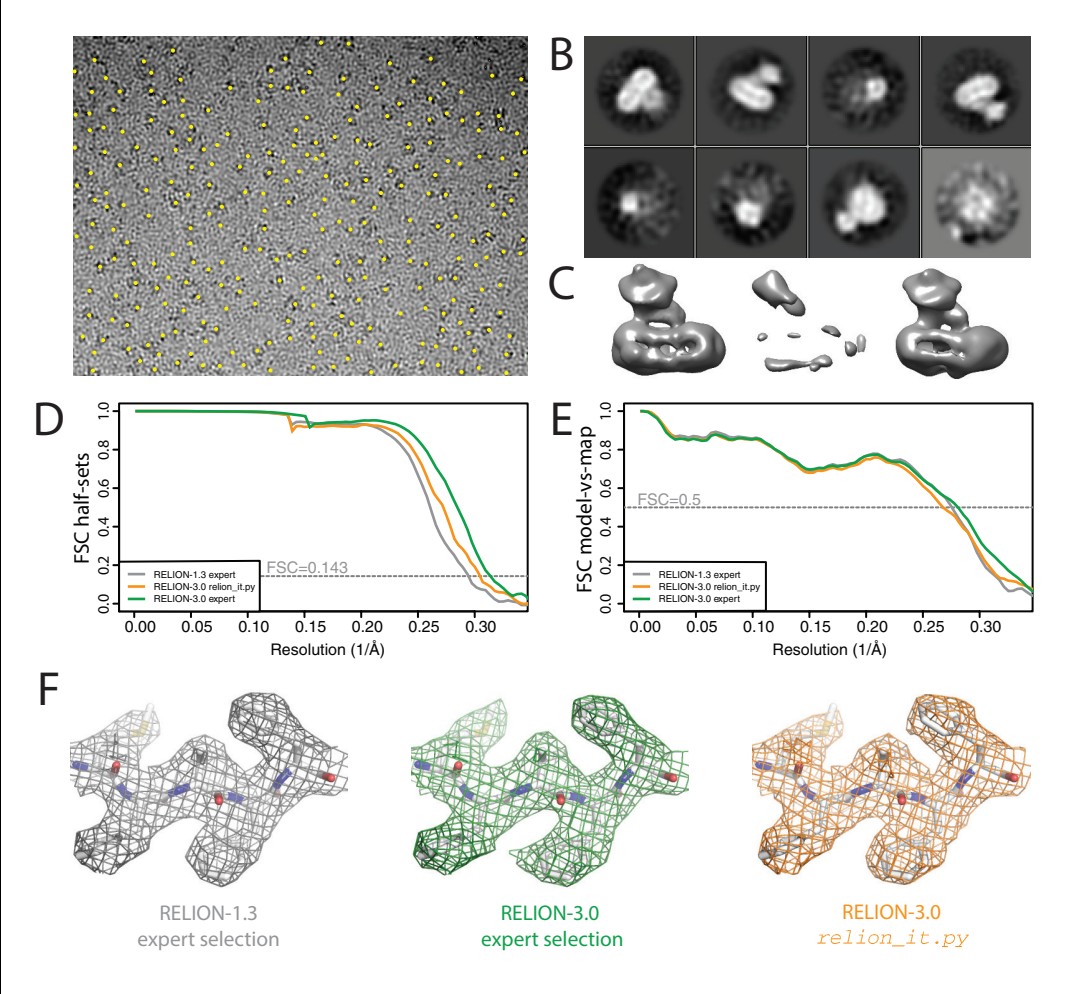

**Figure 6.** Non-interactive data pre-processing. (A) Particle positions as selected by the LoG-based auto-picking algorithm are shown as yellow dots. (B) The 2D class averages for the eight largest classes after LoG-based autopicking of the first batch of 10,000 particles. (C) 3D models generated by the script: after SGD initial model generation (left), and after 3D classification for the largest class (middle) and the second largest class (right). The two maps after 3D classification are thresholded at the same intensity level. (D) FSC curves between independently refined half-maps for EMD-3061 (grey), for the map obtained after non-interactive pre-processing (orange) and for the map obtained after processing of the originally published subset of particles in RELION-3. (E) FSC curves between PDB-5A63 and the same maps as in D. (F) Representative density features for the maps in E and F.

DOI: https://doi.org/10.7554/eLife.42166.011

yielded a map with an estimated resolution of 2.3 Å (*Figure 7A,B*). Re-refining the same data set after only per-particle defocus refinement or beam tilt estimation improved the resolution to 2.2 in both cases. Application of both methods gave 2.1 Å resolution. We then performed Bayesian polishing, which gave 1.94 Å, and classified the resulting shiny particles into six classes. The 134,900 particles that contributed to the three best 3D classes were then re-refined to 1.92 Å resolution. A second round of per-particle defocus refinement and beam tilt estimation, followed by another 3D auto-refinement yielded a final resolution of 1.90 Å. Ewald sphere correction for this data set did not improve the resolution by at least one Fourier shell, although a small increase in FSC values near the estimated resolution was visible. The reconstructed density showed features that one would expect at this resolution (*Figure 7C*). FSC curves between the maps and PDB-5A1A confirmed that RELION-3 improved resolution by 0.3 Å compared to RELION-2.0 (*Figure 7—figure supplement 1*). Application of the `bfactor_plot.py` script produced an estimate for the overall B-factor of 56 Å$^2$

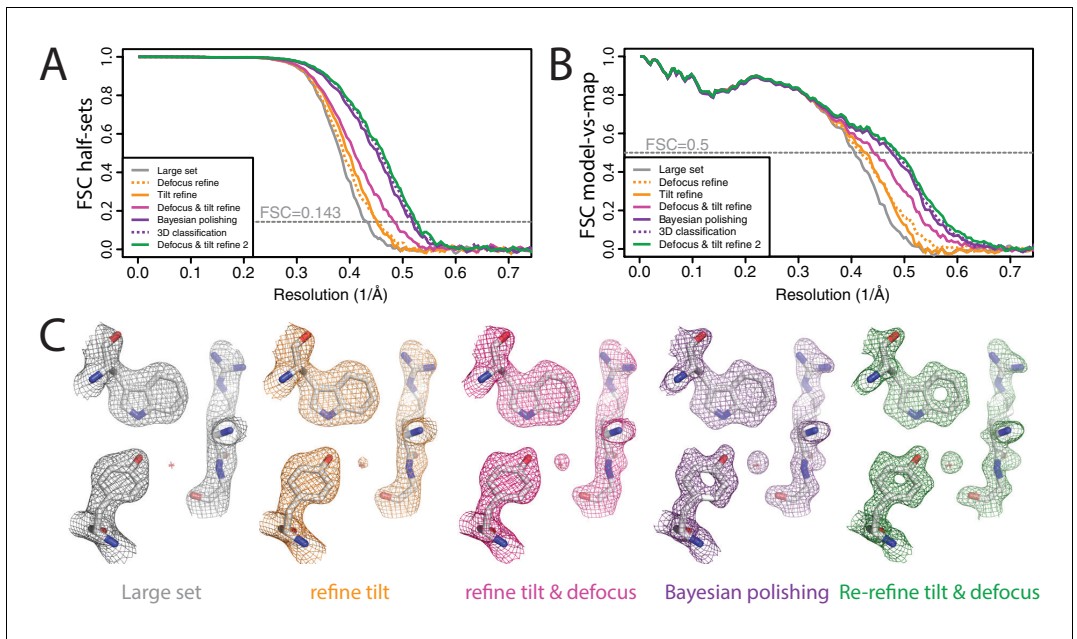

**Figure 7.** High-resolution refinement: $\beta$-galactosidase (**A**) FSC curves between independently refined half-maps for the different stages of processing as explained in the main text. (**B**) FSC curves between PDB-5A1A and the maps at the same stages of processing as in A. (**C**) Representative density features for some of the maps for which FSC curves are shown in A and B.

DOI: https://doi.org/10.7554/eLife.42166.012

The following figure supplements are available for figure 7:

**Figure supplement 1.** FSC curves between PDB-5A1A and EMD-4116, EMD-7770 and the new map reconstructed in RELION-3.

DOI: https://doi.org/10.7554/eLife.42166.013

**Figure supplement 2.** The B-factor plot that was generated automatically by the bfactor_plot.py script.

DOI: https://doi.org/10.7554/eLife.42166.014

---

(*Figure 7—figure supplement 2*). By extrapolation, the script predicts a resolution of 1.75 Å when using four times more particles, and 1.69 Å when using eight times more particles.

Recently, the Subramaniam group also re-processed this data set using new methods for defocus estimation, motion correction and radiation damage weighting (*Bartesaghi et al., 2018*). The resulting map (EMD-7770) was reported as an atomic resolution structure. The resolution according to the 0.143 FSC criterion was 1.9 Å. Despite the same nominal resolution of 1.9 Å, the RELION-3 map correlated with PDB-5A1A to 0.15 Å higher resolution than EMD-7770 (*Figure 7—figure supplement 2*). In *Bartesaghi et al. (2018)* it is argued that the FSC curves may under-estimate the true resolution of cryo-EM maps, and a point is made for showing figures of high-resolution density features to claim atomic resolution instead. We disagree. The high-resolution figures in that paper were made using a map that was sharpened to resolutions well beyond the true resolution, resulting in high levels of noise. This noise was subsequently removed from regions outside the protein density with a high-resolution mask. As this noise still exists within the protein region, the resulting map is prone to incorrect interpretations. FSC curves between half-maps and between model and map are not without their own problems. They do not account for modelling errors or local variations in resolution, and they can be inflated by overfitting or the use of suboptimal masks. Nevertheless, provided suitably soft and low-resolution masks are used and care is taken to avoid overfitting (*Scheres and Chen, 2012*; *Brown et al., 2015*), FSC curves provide an objective overall assessment of map quality. Discarding this information in favour of a few selected density figures would be a mistake.

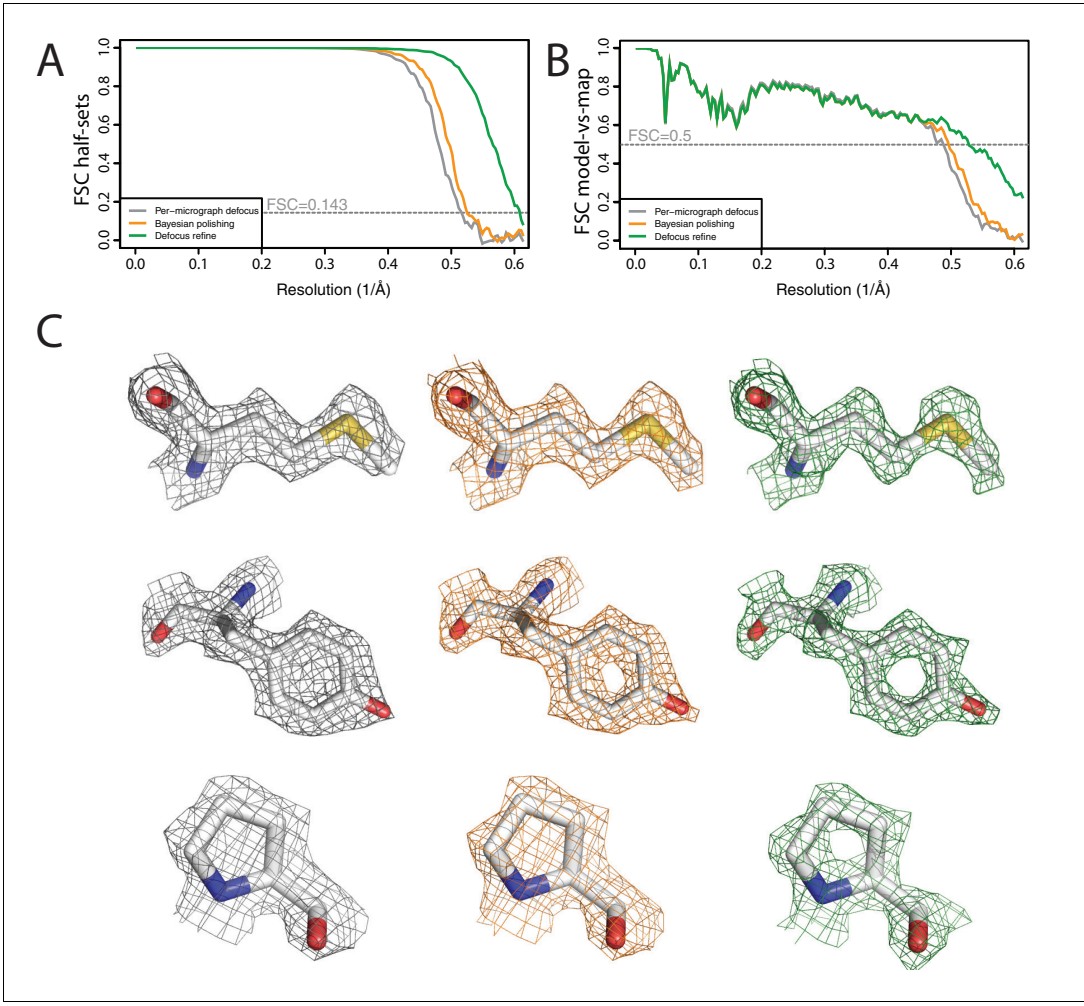

**Figure 8.** High-resolution refinement: apo-ferritin (**A**) FSC curves between independently refined half-maps at different stages of the processing. (**B**) FSC curves between PDB-5N27 (*Ferraro et al., 2017*) and the cryo-EM map at different stages of the processing. (**C**) Representative density features at different stages of the processing.
DOI: https://doi.org/10.7554/eLife.42166.015

The following figure supplement is available for figure 8:

**Figure supplement 1.** The B-factor plot that was generated automatically by the bfactor_plot.py script.
DOI: https://doi.org/10.7554/eLife.42166.016

## High-resolution refinement II: human apo-ferritin

For a final illustration of the combined functionality in RELION-3, we processed a data set on a human apo-ferritin sample, which was a gift from Fei Sun. These data were collected on the same K2 detector and Titan Krios microscope as the horse spleen apo-ferritin data set in section 3.3. We used serialEM to record seven images per hole in the R2/2 Quantifoil grid, before moving the stage to the next hole. Movies with a total dose of 47 electrons per Å$^2$ were fractionated into 40 frames with a pixel size of 0.814 Å. We have deposited these data at EMPIAR-10200.

We again used the relion_it.py script for non-interactive pre-processing of the data. Using default parameters, the script launched motion correction of the 1255 40-frame movies in our own implementation of the MotionCor2 algorithm, and estimated CTF parameters in Gctf (*Zhang, 2016*). LoG-based picking (with $d_{min} = 110$ and $d_{max} = 150$) at the default threshold led to 723,524 particles, which were divided in batches of 10,000 particles. Only the first batch was used for initial model generation and 3D classification. Automated selection of the highest resolution 3D class (at 6.5 Å) yielded a good apo-ferritin model, which was used for template-based auto-picking using the

default (0.4) threshold. The resulting 427,694 particles were used for 2D classification into 200 classes.

We then proceeded processing in an interactive manner. Selection of suitable 2D class averages led to a subset of 426,450 particles, which were subjected to Bayesian polishing and per-particle defocus refinement. A final 3D refinement yielded a resolution of 1.65 Å, which is two resolution shells below the Nyquist frequency (*Figure 8A,B*). Estimation of the beam tilt yielded values below 0.1 mrad, and refinement with these values did not improve the map. As expected at this resolution, the map shows spherical-like features around some of the atoms (*Figure 8C*). However, as these are not yet separated, it would be a stretch to call this an atomic resolution structure. Application of the `bfactor_plot.py` script produced an estimate for the overall B-factor of 66 Å$^2$ (*Figure 8—figure supplement 1*).

Besides the selection of suitable 2D and 3D classes, which as mentioned is a topic of ongoing research in our groups, all of the calculations performed to yield this map could be incorporated into an extended version of the `relion_it.py` script. Thereby, completely automated, high-resolution cryo-EM structure determination of relatively straightforward samples like apo-ferritin is within close reach. This will be useful for the standardised processing of data sets on test samples to assess microscope performance. In the future, improved versions of procedures like this may facilitate high-throughput cryo-EM structure determination, for example for the screening of small-molecule compounds bound to drug targets.

## Conclusions

The new functionality in RELION-3 allows faster, more automated and higher resolution cryo-EM structure determination compared to previous versions. CPU acceleration makes it possible to use a much wider range of hardware efficiently for RELION, including laptops and desktops, and it avoids the box size limits still present when offloading to GPU accelerators. The functionality to schedule and execute jobs from the command-line, coupled with python scripting, provides a versatile and extendable platform to standardise and automate complicated tasks. Corrections for per-particle defocus variations, beam tilt and the Ewald sphere already improve resolutions for currently available data sets, and will become even more important as the resolutions of typical cryo-EM structures improve in the future. RELION is distributed under the GNU General Public License (GPL, version 2), meaning it is completely free to use or modify by anyone, which we hope will maximise its usefulness to the community.

## Acknowledgements

We thank Charles W Congdon from Intel for extensive assistance with implementing the accelerated CPU code; Chris Russo and Richard Henderson for helpful discussions; and Jake Grimmett, Toby Darling and Stefan Fleischmann for help with high-performance computing. This work was funded by the UK Medical Research Council (MC_UP_A025_1013 to SHWS); the Swiss National Science Foundation (SNF: P2BSP2 168735 to JZ); the Japan Society for the Promotion of Science (Overseas Research Fellowship to TN); the Swedish Research Council (2017–04641 to EL) and the Knut and Alice Wallenberg Foundation (to EL).

## Additional information

### Competing interests

Sjors HW Scheres: Reviewing editor, *eLife*. The other authors declare that no competing interests exist.

### Funding

| Funder | Grant reference number | Author |
| --- | --- | --- |
| Schweizerischer Nationalfonds zur Förderung der Wissenschaftlichen Forschung | SNF: P2BSP2 168735 | Jasenko Zivanov |

| | | |
|---|---|---|
| Japan Society for the Promotion of Science | Overseas Research Fellowship | Takanori Nakane |
| Swedish National Infrastructure for Computing | 2017/12-54 | Erik Lindahl |
| Vetenskapsrådet | 2017-04641 | Erik Lindahl |
| Knut och Alice Wallenbergs Stiftelse | | Erik Lindahl |
| Medical Research Council | MC_UP_A025_1013 | Sjors HW Scheres |

The funders had no role in study design, data collection and interpretation, or the decision to submit the work for publication.

### Author contributions

Jasenko Zivanov, Björn O Forsberg, Dari Kimanius, Conceptualization, Software, Formal analysis, Investigation, Methodology, Writing—original draft, Writing—review and editing; Takanori Nakane, Conceptualization, Data curation, Software, Formal analysis, Investigation, Methodology, Writing—original draft, Writing—review and editing; Wim JH Hagen, Conceptualization, Resources, Formal analysis, Investigation, Methodology, Writing—original draft, Writing—review and editing; Erik Lindahl, Conceptualization, Software, Formal analysis, Supervision, Investigation, Writing—original draft, Writing—review and editing; Sjors HW Scheres, Conceptualization, Data curation, Software, Formal analysis, Supervision, Funding acquisition, Validation, Investigation, Visualization, Methodology, Writing—original draft, Project administration, Writing—review and editing

### Author ORCIDs

Jasenko Zivanov (iD) http://orcid.org/0000-0001-8407-0759
Takanori Nakane (iD) http://orcid.org/0000-0003-2697-2767
Wim JH Hagen (iD) http://orcid.org/0000-0001-6229-2692
Erik Lindahl (iD) http://orcid.org/0000-0002-2734-2794
Sjors HW Scheres (iD) http://orcid.org/0000-0002-0462-6540

### Decision letter and Author response

Decision letter https://doi.org/10.7554/eLife.42166.029
Author response https://doi.org/10.7554/eLife.42166.030

## Additional files

### Supplementary files

• Transparent reporting form
DOI: https://doi.org/10.7554/eLife.42166.017

### Data availability

We mostly use publicly available data sets from the EMPIAR data base at EMBL-EBI. For this study, we have submitted to this data base our own data on the human gamma-secretase complex (EMPIAR-10194) and on the high-resolution apo-ferritin sample described in the text (EMPIAR-10200).

The following datasets were generated:

| Author(s) | Year | Dataset title | Dataset URL | Database and Identifier |
|---|---|---|---|---|
| Bai XC, Scheres SHW, Bai XC, Yan C, Yang G, Lu P, Ma D, Sun L, Zhou R, Shi Y | 2018 | An atomic structure of human gamma-secretase [2925 multi-frame micrographs composed of 20 frames each in MRCS format] | https://www.ebi.ac.uk/pdbe/emdb/empiar/entry/10194/ | Electron Microscopy Public Image Archive, EMPIAR-10194 |
| Zivanov J, Nakane T, Hagen WJH, Scheres SHW | 2018 | Human apo-ferritin reconstructed in RELION-3.0 | https://www.ebi.ac.uk/pdbe/emdb/empiar/entry/10200/ | Electron Microscopy Public Image Archive, EMPIAR-10200 |

The following previously published datasets were used:

| Author(s) | Year | Dataset title | Dataset URL | Database and Identifier |
|---|---|---|---|---|
| Bartesaghi A, Merk A, Banerjee S, Matthies D, Wu X, Milne JL, Subramaniam S | 2015 | 2.2 A resolution cryo-EM structure of beta-galactosidase in complex with a cell-permeant inhibitor | https://www.ebi.ac.uk/pdbe/emdb/empiar/entry/10061/ | Electron Microscopy Public Image Archive, EMPIAR-10061 |
| Tan YZ, Lyumkis D | 2017 | 40 Degree Tilted Single-Particle CryoEM of Highly Preferred Orientated Influenza Hemagglutinin Trimer | https://www.ebi.ac.uk/pdbe/emdb/empiar/entry/10097/ | Electron Microscopy Public Image Archive, EMPIAR-10097 |
| Chen DH, Afonine PV, Jakana J, Wang Z, Haase-Pettingell C, Jiang W, Adams PD, King JA, Schmid MF, Chiu W | 2017 | Bacteriophage P22 mature virion capsid protein | https://www.ebi.ac.uk/pdbe/emdb/empiar/entry/10083/ | Electron Microscopy Public Image Archive, EMPIAR-10083 |

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
