## [Decision Letter]

Thank you for submitting your article "RELION-3: new tools for automated high-resolution cryo-EM structure determination" for consideration by *eLife*. Your article has been favorably reviewed by three peer reviewers, including Edward H Egelman as the Reviewing Editor and Reviewer #1, and the evaluation has been overseen by John Kuriyan as the Senior Editor. The following individuals involved in review of your submission have also agreed to reveal their identity: John L Rubinstein (Reviewer #2); Eva Nogales (Reviewer #3).

The reviewers have discussed the reviews with one another and the Reviewing Editor has drafted this decision to help you prepare a revised submission.

Summary:

All reviewers agreed that this paper would be an excellent contribution to the Tools or Resources category in *eLife*, and would be helpful to many people currently using cryo-EM as well as those hoping to use cryo-EM. The new algorithms described in the paper, including CTF/Defocus refinement of individual particles, Ewald sphere correction and beam-tilt correction, appear promising, and it is clear that Relion 3.0 will be an important software package for many people. No major concerns were raised, only minor points that can be easily addressed.

Minor points:

1) The authors estimate the resolution in some instances to a hundredth of an Å. This level of precision makes no sense in cryo-EM, where the FSC curve is never monotonic, and results are easily influenced by masking. In addition, Sjors recently wrote in a CCPEM forum that "we have seen artificially high FSC curves in cases of strong 'rotational smearing' of the particles due to extremely broad posterior probability functions of the orientations." So simply smearing the map can improve the FSC, as previously noted by Grigorieff. Thus, defining the FSC resolution, which can be easily altered by tenths of an Å by masking and smearing, to a precision of a hundredth of an Å defies common sense. Therefore, 3.49 should be 3.5, and 3.11 should be 3.1.

2) Figure 1 needs the y-axes to be labelled in both parts. Without knowing precisely what is being plotted, it is confusing that the function oscillates about zero (I had guessed correlation, but that would be positive everywhere there is correlation).

3) CTF refinement: It might be useful to add a few statements regarding the performance of the per-particle CTF refinement algorithm when dealing with i) particles of different size (e.g. are smaller particles less reliably fit), ii) reconstructions of different resolution (e.g. will using a low-resolution reference structure lead to degraded defocus estimates; if so, at which resolutions has this been observed?), and iii) micrographs with different numbers of particles. Regarding the last point, it appears that the algorithm does not use the particle population of a micrograph to improve the accuracy of CTF fitting, e.g. by fitting a plane or to avoid outliers (e.g. Figure 4, supplement 1). This would imply that the number of particles per micrograph does not play a role in the per-particle use case – is this correct?

Also, since the refined map is needed for CTF refinement, how reliable is the procedure with reconstructions at medium resolution (in the 5-7Å regime)? Finally, have the authors compared their method with the local CTF estimation method done using Gctf, where no input map is needed?

4) Ewald sphere correction: Could the authors give the radius of the test specimen used, and the resolutions at which Ewald sphere curvature becomes limiting for this radius, according to previous estimates in the literature (e.g. by DeRosier, Grigorieff, and others)?

5) Automated data exploration: The script failed to select the correct structure because it used, at equal resolution, the largest class. Did any of the other refinement parameters point to the correct structure (e.g. translation or rotation accuracies from the alignment)?

6) Β-galactosidase: The comparison to previous reconstructions from the same and other groups is insightful and the emphasis on using proper validation statistics is important.

---

## [Author Response]

Minor points:1) The authors estimate the resolution in some instances to a hundredth of an Å. This level of precision makes no sense in cryo-EM, where the FSC curve is never monotonic, and results are easily influenced by masking. In addition, Sjors recently wrote in a CCPEM forum that "we have seen artificially high FSC curves in cases of strong 'rotational smearing' of the particles due to extremely broad posterior probability functions of the orientations." So simply smearing the map can improve the FSC, as previously noted by Grigorieff. Thus, defining the FSC resolution, which can be easily altered by tenths of an Å by masking and smearing, to a precision of a hundredth of an Å defies common sense. Therefore, 3.49 should be 3.5, and 3.11 should be 3.1.

Leaving such precise numbers for the lower resolutions was an oversight, which we have corrected in the revised version. Note that we have left the precise values for resolutions beyond 2 Angstroms, where improvements in the third digit do become relevant.

2) Figure 1 needs the y-axes to be labelled in both parts. Without knowing precisely what is being plotted, it is confusing that the function oscillates about zero (I had guessed correlation, but that would be positive everywhere there is correlation).

We have added labels to the Y-axes in Figure 1.

3) CTF refinement: It might be useful to add a few statements regarding the performance of the per-particle CTF refinement algorithm when dealing with i) particles of different size (e.g. are smaller particles less reliably fit).

We have not observed any issues with the defocus estimation of small particles, although this is not described in the paper. It could be that their orientation determination (with five parameters) is more difficult than their defocus determination (with only a single parameter).

ii) Reconstructions of different resolution (e.g. will using a low-resolution reference structure lead to degraded defocus estimates; if so, at which resolutions has this been observed?).

The higher the resolution in the reference, the more information in principle is available to estimate the defocus. Our hemagglutinin example shows that at least 6.6A does not seem to be a problem. We added the following sentence: "This example demonstrates that even for models with intermediate resolution (in this case 6.6 Å) our per-particle defocus refinement can yield improved defoci."

iii) micrographs with different numbers of particles. Regarding the last point, it appears that the algorithm does not use the particle population of a micrograph to improve the accuracy of CTF fitting, e.g. by fitting a plane or to avoid outliers (e.g. Figure 4, supplement 1). This would imply that the number of particles per micrograph does not play a role in the per-particle use case – is this correct?

This is correct. We have modified the following sentence: "The expression in Equation 3 allows the CTF-parameters θ = {δ0, δA, φA, Cs, χ} to be determined either as constant for all particles in an entire micrograph or for every particle separately. In the latter case, the second sum consists of one single term corresponding to the index p of the particle in question, and the result will be independent on the number of particles per micrograph."

Also, since the refined map is needed for CTF refinement, how reliable is the procedure with reconstructions at medium resolution (in the 5-7Å regime)? Finally, have the authors compared their method with the local CTF estimation method done using Gctf, where no input map is needed?

We haven't. The idea, as per Figure 1, is that one would run gCTF before refining the defoci in RELION. The text reads: "Therefore, in practice it would probably be faster to start from defoci provided by local CTF estimation programs, such as ctftilt [Mindell and Grigorieff, 2003], gctf [Zhang, 2016] or Warp [Tegunov and Cramer, 2018]."

4) Ewald sphere correction: Could the authors give the radius of the test specimen used, and the resolutions at which Ewald sphere curvature becomes limiting for this radius, according to previous estimates in the literature (e.g. by DeRosier, Grigorieff, and others)?

We have added the following text: "DeRosier, [2000] estimates that the frequency *k** for which Ewald sphere curvature becomes limiting for a particle with diameter d is approximated by:

k*=1.4dλ.

And: "The diameter of the P22 virion is approximately 700Å, which leads to an estimated resolution limit of 3.1Å due to Ewald sphere curvature [DeRosier, 2000]."

5) Automated data exploration: The script failed to select the correct structure because it used, at equal resolution, the largest class. Did any of the other refinement parameters point to the correct structure (e.g. translation or rotation accuracies from the alignment)?

We have implemented the script to make choices on what we thought was the optimal procedure for general applicability: i.e. to select classes based on resolution, and in case of equal resolution, on class size. We considered it a better reflection of the power of the current implementation to openly describe its limitations, rather than tuning it to this specific case and getting a non- representative (overly optimistic) result. We do not expect that merely changing resolution for accuracy in alignment as a decision criterion will enhance the general applicability of the script. As mentioned in the manuscript, we are currently investigating how to improve the automated procedures, and we expect that the actual script will change considerably in the future. The main advance in implementation is the scriptable interface, which allows such developments to happen independently from the RELION source code, and therefore also easily by others.

6) Β-galactosidase: The comparison to previous reconstructions from the same and other groups is insightful and the emphasis on using proper validation statistics is important.

Your support is appreciated. Although we try to avoid direct comparisons with work from others in general, we felt that we needed to warn against suboptimal procedures in this case.